# Pedestrian Attribute Recognition via Hierarchical Cross-Modality HyperGraph Learning

## Abstract

Current Pedestrian Attribute Recognition (PAR) algorithms typically focus on mapping visual features to semantic labels or attempt to enhance learning by fusing visual and attribute information. However, these methods fail to fully exploit attribute knowledge and contextual information for more accurate recognition. Although recent works have started to consider using attribute text as additional input to enhance the association between visual and semantic information, these methods are still in their infancy. To address the above challenges, this paper proposes the construction of a multi-modal knowledge graph, which is utilized to mine the relationships between local visual features and text, as well as the relationships between attributes and extensive visual context samples. Specifically, we propose an effective multi-modal knowledge graph construction method that fully considers the relationships among attributes and the relationships between attributes and vision tokens. To effectively model these relationships, this paper introduces a knowledge graph-guided cross-modal hypergraph learning framework to enhance the standard pedestrian attribute recognition framework. Comprehensive experiments on multiple PAR benchmark datasets have thoroughly demonstrated the effectiveness of our proposed knowledge graph for the PAR task, establishing a strong foundation for knowledge-guided pedestrian attribute recognition. The source code of this paper will be released upon acceptance.

## 1 Introduction

Pedestrian Attribute Recognition (PAR) targets to predict the human attributes like *long hair*, *long pants* from a given attribute set based on a given pedestrian image. It can be seen as a middle-level semantic representation and contributes to other computer vision tasks like person re-identification (Lin et al., 2019), pedestrian detection (Zhang et al., 2020), object tracking (Li et al., 2024), and text-based person retrieval (Aggarwal et al., 2020). PAR has been widely deployed in practical smart video surveillance, autonomous driving, etc. However, the performance of pedestrian attribute recognition is still poor in challenging scenarios, e.g., low illumination, motion blur, and occlusion.

Current algorithms usually formulate the PAR as a multi-label classification problem and learn a mapping function from the input image to the semantic labels using the encoder-decoder framework. Specifically, DeepMAR (Li et al., 2015a) proposed by Li et al. first learning a deep neural network and predicts the pedestrian attributes in an end-to-end manner. Later, some researchers further enhanced such a framework by fusing the vision and attribute features, e.g., VTB (Cheng et al., 2022), SeqPAR (Jin et al., 2025a), and PromptPAR (Wang et al., 2024a), as shown in Fig. 1 (a, b). Specifically, Cheng et al. first extract the vision and attribute features using the Transformer network and fuse them for high-performance PAR. After that, Wang et al. propose to enhance this framework by prompting the pre-trained multi-modal foundation model CLIP (Radford et al., 2021), Jin et al. (Jin et al., 2025a) formulate the PAR as a sequence generation problem based on a Transformer network.

Despite significant progress having been made with the help of Transformer networks and foundation models, the PAR still suffers from the following issues: 1). Existing works usually model the PAR as a mapping from the given pedestrian image to the semantic labels using CNN, LSTM, or Transformers, but ignore the mining of semantic information of pedestrian attributes. Obviously,

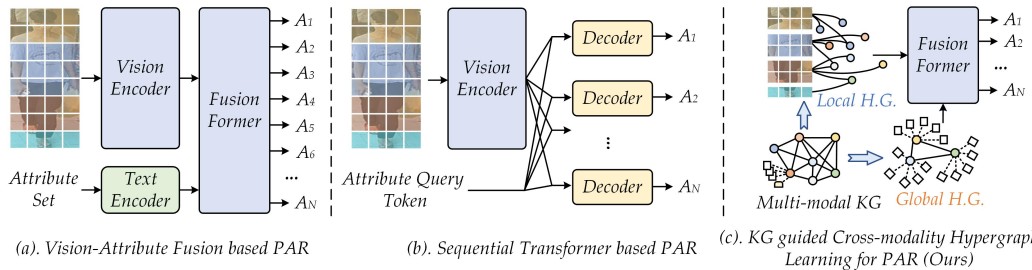

(a). Vision-Attribute Fusion based PAR    (b). Sequential Transformer based PAR    (c). KG guided Cross-modality Hypergraph Learning for PAR (Ours)

Figure 1: Comparison between existing (a). vision-text fusion based PAR, (b). sequential Transformer based PAR, and (c). our newly proposed knowledge graph guided cross-modality hypergraph learning for PAR.

the semantic gaps will hinder the further improvement of these models. 2). Some models attempt to fuse the semantic attribute phases into the vision feature learning stage, but few of them consider the higher-order relations between the attributes and vision features. Thus, it is natural to raise the following question: "*How can we exploit the higher-order relations between the pedestrian attributes and the attribute-vision features for high-performance pedestrian attribute recognition?*"

Inspired by the success of knowledge graph (Liang et al., 2024), in this paper, we attempt to bridge the semantic gap between the vision and attribute information, and achieve knowledge graph guided hierarchical cross-modal hypergraph learning for pedestrian attribute recognition. The key insight of this paper is to build a multi-modal knowledge graph that takes the human body, attributes as the entity, and trunk-attribute as the relation. It also takes the language captions of each attribute and the context vision samples as attributes of each entity to help the models better understand the attributes. As shown in Fig. 2, we enhance the standard visual-attribute mapping framework from two perspectives through the constructed multi-modal knowledge graph, namely *local patch-attribute relation mining* and *attribute-global context sampling relation mining*. We adopt the hypergraph to capture the higher-order relations in these two modules and encode them using the LA-UniGNN and AG-UniGNN networks. After that, we concatenate these processed tokens and feed them into a multi-modal Transformer for vision-semantic aggregation. Finally, we adopt a prediction head (i.e., the Feed-Forward Network, FFN) to output the predicted pedestrian attributes.

To sum up, the main contributions of this paper can be summarized as the following three aspects:

• We propose a novel multi-modal pedestrian attribute knowledge graph, termed M2PA-KG, this knowlege graph is the first large-scale multi-modal knowledge graph in the PAR domain, containing comprehensive attribute entities, human body entities, attribute descriptions, and visual context samples.

• We propose a novel Hierarchical Cross-Modal HyperGraph Learning for Knowledge Graph Augmented Pedestrian Attribute Recognition, termed KGPAR. This framework fully exploits the high-order relationships between attributes and between attributes and images, achieving knowledge-guided high-performance pedestrian attribute recognition through effective hypergraph modeling.

• Extensive experiments on multiple PAR benchmark datasets fully validated the effectiveness of our proposed M2PA-KG for the PAR task, laying a solid foundation for knowledge-guided pedestrian attribute recognition.

## 2 RELATED WORKS

### 2.1 PEDESTRIAN ATTRIBUTE RECOGNITION

Pedestrian Attribute Recognition (PAR) aims to classify pedestrians based on attributes such as gender and clothing.Par tasks can be divided into these categories, CNN-based methods, attention- and pose-guided methods, and transformer-based methods. CNN-based methods were the earliest to dominate PAR research. These methods use convolutional neural networks such as VGG and ResNet to extract global or local visual features from pedestrian images, followed by multi-

label classifiers to predict attributes. Abdulnabi et al. (Abdulnabi et al., 2015) employ CNNs for pedestrian attribute analysis and propose a multi-task learning strategy that uses multiple CNNs to learn attribute-specific features, enabling knowledge sharing across networks.Diba et al. (Diba et al., 2016) design an iterative clustering algorithm for pedestrian attribute recognition, referred to as Deep-CAMP, which progressively refines attribute-specific clusters to improve recognition performance. Later, in order to improve the spatial alignment between attributes and corresponding body regions, attention guidance and posture guidance methods were proposed. For example, HP-Net (Liu et al., 2017) combines multi-scale attention mechanisms across different semantic levels to optimize feature representation. VeSPA (Sarafianos et al., 2018) uses the view sensitive attention mechanism to adaptively select the area Er according to the pedestrian's perspective networks.

Transformers like ViT improved long-range dependency modeling but still encounter difficulties in capturing subtle attribute variations and addressing attribute imbalance, as demonstrated in ExpIB-Net (Wu et al., 2023) and SOFAFormer (Wu et al., 2024).Recent works leverage multimodal learning for enhanced recognition.VTB (Cheng et al., 2022) combined visual and textual features through transformer architectures, and LLM-PAR (Jin et al., 2025b) incorporated large language models to refine semantic reasoning. But these methods still face challenges in cross-dataset generalization.

## 2.2 KNOWLEDGE GRAPH

In recent years, Knowledge Graphs (KGs) have been widely applied across various domains due to their ability to represent complex relationships and semantic information. For instance, Google Knowledge Graph (Singhal et al., 2012) enhances search accuracy through semantic understanding; IBM Watson Health (Liang et al., 2019) uses KGs to support personalized treatment recommendations in healthcare; and SR-GNN (Wu et al., 2019) in e-commerce improves recommendation systems by modeling user behavior and product attributes. Knowledge graphs have also found applications in finance, education, and other fields, demonstrating their versatile utility.

In this study, we construct a multi-modal knowledge graph to provide structured guidance for subsequent hypergraph construction. By integrating multiple modalities such as images and text, the knowledge graph effectively captures the relationships between entities, offering a solid foundation for hypergraph modeling. This multi-modal knowledge graph serves as a powerful semantic support for downstream tasks, contributing significantly to the optimization of pedestrian attribute recognition.

## 3 METHODOLOGY

### 3.1 OVERVIEW

The overall framework of our architecture is illustrated in Fig. 2. Our method first obtains input embeddings through CLIP §3.2. and constructs a multi-modal knowledge graph to capture semantic and structural relationships among visual attributes §3.3. To further model both the internal topological structure of individual images and the semantic correlations across images, we design a hierarchical cross-modal hypergraph learning mechanism §3.4. Finally, the learned representations are integrated for attribute prediction §3.5.

### 3.2 INPUT EMBEDDING

In this work, we adopt the image encoder $f(\cdot)$ and text encoder $g(\cdot)$ from CLIP as feature extractors for vision and language, respectively. Formally, given an image $x^v$, we first apply a patch embedding layer $\text{PatchEmbed}(\cdot)$ to split the input image $x^v$ and project it into fixed-size patch embeddings $v_p = [v_p^1, v_p^2, \ldots, v_p^N] \in \mathbb{R}^{N \times d}$, where $N$ is the number of image patches, and $d$ is the feature dimension.

To model local region semantics, the input image is horizontally divided into $R$ regions (e.g., head, upper, lower, foot), and a corresponding learnable $d$-dimensional token $\mathbf{CLS}_{l_i}$ is assigned to each region. In addition, following existing methods, we introduce a learnable $d$-dimensional global classification token $\mathbf{CLS_g}$. Therefore, we get the complete input sequence $v_{all} = [\mathbf{CLS_g}; \ \mathbf{CLS}_{l_1}, \mathbf{CLS}_{l_2}, \ldots, \mathbf{CLS}_{l_R}; \ v_p]$. Feeding this sequence into the image encoder

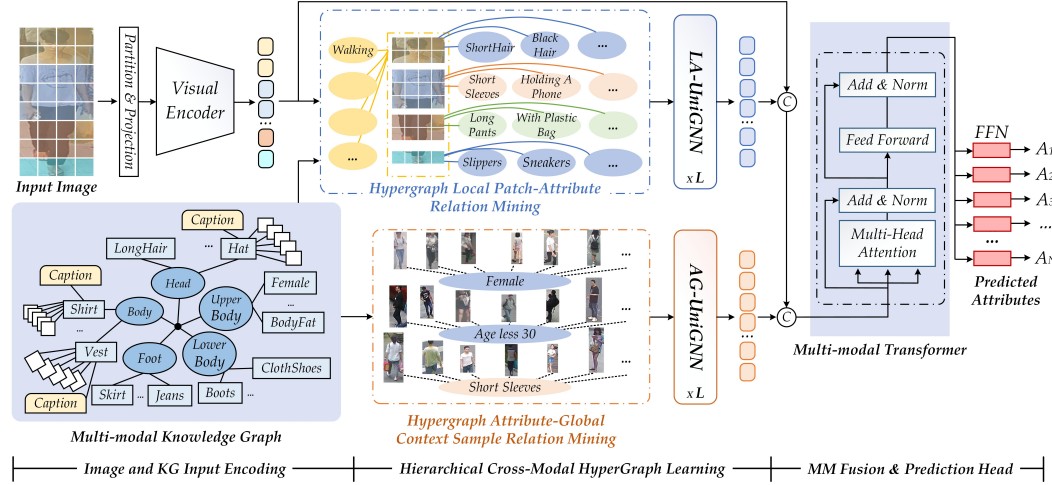

Figure 2: An overview of our proposed knowledge graph augmented PAR by learning the hierarchical cross-modal hypergraph.

$f(\cdot)$ to get the final input embeddings:

$$[c_g; c_{l_1}, \ldots, c_{l_R}; h_p] = f(v_{all}), \tag{1}$$

where $c_g$, $c_{l_i}$, and $h_p$ are the global embedding, local embeddings, and the patch-level embeddings, respectively. These embeddings are in a shared embedding space enabling alignment with text modalities and facilitating downstream tasks.

### 3.3 MULTI-MODAL KNOWLEDGE GRAPH CONSTRUCTION

To capture the semantic and structural relationships among attributes, we construct a multi-modal knowledge graph for each dataset, as shown in Fig. 4 in the appendix. This knowledge graph serves as the foundation for subsequent learning tasks. Formally, the knowledge graph is defined as:

$$G = (V, E, S, I) \tag{2}$$

where $V$, $E$, $S$, and $I$ represent the set of nodes, edges, textual features, and visual features, respectively. Each pedestrian attribute is modeled as a node and systematically organized into five semantic modules: head, body, upper body, lower body, and foot. Every node is associated with both textual and visual representations, while edges are established according to the co-occurrence patterns of attributes observed in the dataset.

Specifically, we construct an attribute co-occurrence adjacency matrix $A \in \mathbb{R}^{M \times M}$ by aggregating attribute labels across all images:

$$A_{ij} = \sum_{k=1}^{N} L_{ki} \cdot L_{kj}, \tag{3}$$

where $L \in \{0, 1\}^{N \times M}$ denotes the label matrix, and $A_{ii}$ indicates the frequency of attribute $i$. This adjacency matrix serves to establish edges between nodes in the knowledge graph. To facilitate the propagation of multimodal features, we perform row-wise normalization of the adjacency matrix:

$$\tilde{A}_{ij} = \frac{A_{ij}}{A_{ii}}, \quad A_{ii} > 0. \tag{4}$$

Each element $\tilde{A}_{ij}$ of the normalized matrix is employed as the edge weight between nodes $i$ and $j$, reflecting the strength of their association.

This knowledge graph not only preserves the semantic and visual correlations among attributes but also provides explicit guidance for the construction of local and global hypergraphs, thereby promoting efficient interaction and propagation of multimodal features.

## 3.4 HIERARCHICAL CROSS-MODAL HYPERGRAPH LEARNING

To capture the internal topological structure of individual images and the semantic relationships across images, we design a hierarchical cross-modal hypergraph learning mechanism. Within this mechanism, we construct both local and global hypergraphs to effectively model intra-image feature interactions and inter-image correlations.

• **Local HyperGraph** As demonstrated in Section 3.3, each pedestrian image is first divided into several predefined regions $R = \{\text{body}, \text{head}, \text{upper}, \text{lower}, \text{foot}\}$. To establish semantic correspondences between text $T_r = \{t_1, \ldots, t_{M_r}\}$ and visual $h_{p_r} = \{h_{p_1}, \ldots, h_{p_{N_r}}\}$ within each region $r \in R$, we compute the similarity matrix:

$$S_r = \mathbf{h}_{p_r} \cdot T_r^\top \in \mathbb{R}^{N_r \times M_r}. \tag{5}$$

For each textual token $\mathbf{t}_j \in T_r$, only the visual patch tokens $\mathbf{h}_{p_i} \in \mathbf{h}_p$ whose similarity exceeds a threshold $\tau$ are retained. These selected patch tokens are then combined with the corresponding textual token to form a hyperedge, thereby constructing a region-level hypergraph.

Finally, all regional hypergraph nodes and hyperedges are merged to form the complete local hypergraph. This hypergraph is then input into the UniGNNHuang & Yang (2021) for encoding, producing region features $\mathbf{H}_{\text{local}}$ that capture the fused text-visual semantic relationships.

• **Global HyperGraph** To jointly model the semantic alignment between images and attribute texts, we construct a Global HyperGraph. The attribute texts are mapped into a shared feature space using the CLIP text encoder, yielding the text embedding matrix $F_{\text{text}} = [f_1^{\text{text}}, f_2^{\text{text}}, \ldots, f_M^{\text{text}}]$, where $d$ denotes the embedding dimension. For the image collection, global class tokens extracted by the CLIP visual encoder serve as semantic representations, forming the image embedding matrix $F_{\text{img}} = [f_1^{\text{img}}, f_2^{\text{img}}, \ldots, f_N^{\text{img}}]$. By concatenating image and text features, we obtain the node feature matrix of the global hypergraph

$$F = \begin{bmatrix} F_{\text{img}} \\ F_{\text{text}} \end{bmatrix} \in \mathbb{R}^{(N+M) \times d}. \tag{6}$$

To capture the image–text correspondence, an image-to-attribute association matrix $Y_{\text{img}} \in \mathbb{R}^{N \times M}$ is constructed, where

$$Y_{\text{img}}[i, j] = \begin{cases} 1, & \text{if image } I_i \text{ is associated with attribute } t_j, \\ 0, & \text{otherwise.} \end{cases} \tag{7}$$

To maintain the independence of text nodes, an identity matrix $Y_{\text{text}} = I_M \in \mathbb{R}^{M \times M}$ is introduced. Consequently, the association matrix $A$ of the global hypergraph is constructed as follows:

$$A = \begin{bmatrix} Y_{\text{img}} \\ Y_{\text{text}} \end{bmatrix} \in \mathbb{R}^{(N+M) \times M}. \tag{8}$$

The constructed global hypergraph is then input into UniGNNHuang & Yang (2021) for encoding, generating image-text aligned representations $\mathbf{H}_{\text{global}}$ that capture global semantic relationships.

## 3.5 ATTRIBUTES PREDICTION

Before performing prediction, we apply a multi-modal fusion strategy to integrate the visual features derived from both the global and local hypergraphs.

• **Multi-modal Fusion** Inspired by VTB (Cheng et al., 2022), we concatenate the visual features with the constructed global hypergraph and local hypergraph, and feed them into a multi-modal Transformer for joint modeling and deep interaction across different modalities. Specifically, the concatenated input sequence is:

$$\mathbf{Z} = [\mathbf{H}_{\text{local}}; \mathbf{H}_{\text{global}}; f(v_{\text{all}})]. \tag{9}$$

The multi-head self-attention mechanism enables the Transformer to capture high-order dependencies across modalities while integrating structural information at different granularities. After this

fusion, the model obtains more discriminative multi-modal representations, which ultimately enhance attribute prediction.

• **Prediction** Based on the representations obtained from the Multi-modal Fusion, we employ a feed-forward network (FFN) as the classifier to regress attribute scores:

$$R = \text{FFN}(Z) = \sigma(wZ + b) \tag{10}$$

where $w$ and $b$ denote the weight and bias of the FFN, and $\sigma(\cdot)$ is the Sigmoid activation function for producing the final attribute probabilities.

• **Loss Computation** In this study, we employ a loss function composed of two components to optimize the training process. The first is the Global–Local Similarity Loss ($L_{GL}$), which models the similarity between the global CLS token, local region CLS tokens, and attribute representations, and aligns them with the ground truth. This loss effectively evaluates the semantic consistency of visual features at both global and local levels. It is formulated as a binary cross-entropy function:

$$L_{GL} = -\frac{1}{M} \sum_{i=1}^{M} \sum_{j=1}^{N} \left( y_{ij} \log(p_{ij}^{GL}) + (1 - y_{ij}) \log(1 - p_{ij}^{GL}) \right) \tag{11}$$

where $p_{ij}^{GL}$ denotes the similarity prediction of the $i$-th sample for the $j$-th attribute, and $y_{ij}$ represents the corresponding ground truth label. This loss ensures that attribute embeddings achieve semantic alignment at both global and local levels. The second is the Weighted Cross-Entropy Loss ($L_{CLS}$), which addresses the class imbalance commonly encountered in pedestrian attribute recognition. The weight for each attribute class is determined according to its frequency in the training set, defined as $w_j = e^{-r_j}$, where $r_j$ denotes the occurrence ratio of the $j$-th attribute. The loss is defined as:

$$L_{CLS} = -\frac{1}{M} \sum_{i=1}^{M} \sum_{j=1}^{N} w_j \left( y_{ij} \log(p_{ij}) + (1 - y_{ij}) \log(1 - p_{ij}) \right) \tag{12}$$

where $p_{ij}$ indicates the predicted probability of the $j$-th attribute for the $i$-th image. By incorporating this weighting scheme, the loss function reduces the dominance of majority classes in the optimization process, thereby enhancing the recognition capability for minority attributes. The final overall optimization objective is formulated as:

$$L = L_{CLS} + \alpha L_{GL} \tag{13}$$

where $\alpha$ is a trade-off parameter that balances classification performance and global–local alignment.

## 4 EXPERIMENTS

### 4.1 DATASETS AND EVALUATION METRIC

To demonstrate the effectiveness of our proposed method, we conduct experiments on five PAR benchmark datasets, including standard PETA (Deng et al., 2014), PA100K (Liu et al., 2017), RAPv1 (Li et al., 2016), RAPV2 (Li et al., 2018), MSP60K (Jin et al., 2025b). We adopt five widely used evaluation metrics to rigorously assess model performance: mean Accuracy (mA), Accuracy (Acc), Precision (Prec), Recall, and F1-score (F1). Some experiments are provided in the supplementary materials.

### 4.2 IMPLEMENTATION DETAILS

During the training phase, we employ the ViT-L/14 version of CLIP as the visual encoder, with input images resized to $224 \times 224$. The model is trained with a batch size of 16 for a total of 100 epochs. This training configuration is consistently applied across the RAPv1, RAPv2, PETA, and PA100K datasets. The initial learning rate is set to 7e-4 and decays progressively at a rate of 1e-4, while an additional 0.01 learning rate decay is applied to stabilize the training process. Parameter updates are performed using the AdamW optimizer. Further implementation details are available in our source code.

### 4.3 COMPARISON ON PUBLIC BENCHMARKS

• **Results on PETA (Deng et al., 2014) dataset.** Our method achieves 88.50, 83.27, 89.74, 89.56, and 89.43 in mA, Accuracy, Precision, Recall, and F1, respectively. Although it does not surpass methods such as DRFormer or OAGCN on individual metrics, the overall performance remains consistently high, demonstrating a well-balanced capability across metrics.

• **Results on PA100K (Liu et al., 2017) dataset.** Our approach reaches 87.95, 84.01, 89.55, 91.65, and 90.28 in mA, Accuracy, Precision, Recall, and F1, respectively. Compared with FRDL, it achieves improvements of 1.65 and 2.36 on mA and F1, respectively; it also outperforms PARformer in terms of Accuracy and F1.

• **Results on RAPv1 (Li et al., 2016) dataset.** Our method obtains 85.46, 71.81, 80.54, 85.16, and 82.46 in mA, Accuracy, Precision, Recall, and F1, respectively. Compared with OAGCN, it achieves comparable or even superior results without relying on additional viewpoint information.

• **Results on RAPv2 (Li et al., 2018) dataset.** The results are 83.02, 70.03, 78.33, 85.17, and 81.27 in mA, Accuracy, Precision, Recall, and F1, respectively. Although Accuracy is slightly lower than some other methods, the performance in mA and F1 remains competitive.

• **Results on MSP60K (Jin et al., 2025b) datasets.** We evaluated our method on the MSP60K dataset, and the experimental results are shown in Table 3. In the random split, our method achieved mA 79.03, Acc 76.94, Precision 84.14, Recall 88.14, and F1 85.69. In the cross-domain split, our method achieved mA 63.82, Acc 53.67, Precision 65.86, Recall 72.35, and F1 68.41, demonstrating stable performance. These results demonstrate that our method exhibits excellent performance on the MSP60K dataset, with well-balanced and strong capabilities across multiple key metrics, validating its effectiveness and stability in cross-domain pedestrian attribute recognition tasks.

Table 1: Comparison with state-of-the-art methods on PETA and PA100K datasets. The best and second best results are highlighted in **bold** and underlined, respectively. Missing values are marked with "-".

| Methods | Publish | PETA | | | | | PA100K | | | | |
|---|---|---|---|---|---|---|---|---|---|---|---|
| | | mA | Acc | Prec | Rec | F1 | mA | Acc | Prec | Rec | F1 |
| SSCNet (Jia et al., 2021a) | ICCV21 | 86.52 | 78.95 | 86.02 | 87.12 | 86.99 | 81.87 | 78.89 | 85.98 | 89.10 | 86.87 |
| CAS (Yang et al., 2021) | IJCV21 | 86.40 | 79.93 | 87.03 | 87.33 | 87.18 | 82.86 | 79.64 | 86.81 | 87.79 | 85.18 |
| IAA (Wu et al., 2022) | PR22 | 85.27 | 78.04 | 86.08 | 85.80 | 85.64 | 81.94 | 80.31 | 88.36 | 88.01 | 87.80 |
| DRFormer (Tang & Huang, 2022) | NC22 | 89.96 | 81.30 | 85.68 | 91.08 | 88.30 | 82.47 | 80.27 | 87.60 | 88.49 | 88.04 |
| VAC (Guo et al., 2022) | IJCV22 | - | - | - | - | - | 82.19 | 80.66 | 88.72 | 88.10 | 88.41 |
| DAFL (Jia et al., 2022) | AAAI22 | 87.07 | 78.88 | 85.78 | 87.03 | 86.40 | 83.54 | 80.13 | 87.01 | 89.19 | 88.09 |
| VTB (Cheng et al., 2022) | TCSVT22 | 85.31 | 79.60 | 86.76 | 87.17 | 86.71 | 83.72 | 80.89 | 87.88 | 89.30 | 88.21 |
| PARformer (Fan et al., 2023) | TCSVT23 | 89.32 | 82.86 | 88.06 | **91.98** | 89.06 | 84.46 | 81.13 | 88.09 | 91.67 | 88.52 |
| OAGCN (Lu et al., 2023) | TMM23 | **89.91** | 82.95 | 88.26 | 89.10 | 88.68 | 83.74 | 80.38 | 84.55 | 90.42 | 87.39 |
| SSPNet (Shen et al., 2024) | PR24 | 88.73 | 82.80 | 88.48 | 90.55 | 89.50 | 83.58 | 80.63 | 87.79 | 89.32 | 88.55 |
| SOFA (Wu et al., 2024) | AAAI24 | 87.10 | 81.10 | 87.80 | 88.40 | 87.80 | 83.40 | 81.10 | 88.40 | 89.00 | 88.30 |
| FRDL (Zhou et al., 2024) | ICML24 | 88.59 | - | - | - | 89.03 | **89.44** | - | - | - | 88.05 |
| PromptPAR (Wang et al., 2024a) | TCSVT24 | 88.76 | 82.84 | 89.04 | 89.74 | 89.18 | 87.47 | 83.78 | 89.27 | 91.70 | 90.15 |
| SequencePAR (Jin et al., 2025a) | PR25 | - | **84.92** | **90.44** | 90.73 | **90.46** | - | 83.94 | **90.38** | 90.23 | 90.10 |
| EVSITP (Wu et al., 2025a) | CVPR25 | 89.65 | 83.93 | 89.67 | 90.73 | 90.20 | 88.66 | **84.54** | 89.90 | **92.09** | **90.98** |
| HDFL Wu et al. (2025b) | Neural Network25 | 87.55 | 79.66 | 87.08 | 87.16 | 86.85 | 84.92 | 80.23 | 87.45 | 88.74 | 87.72 |
| FOCUS (An et al., 2025) | ICME25 | 88.04 | 81.96 | 88.56 | 89.07 | 88.54 | 83.90 | 81.23 | 89.29 | 88.97 | 88.41 |
| KGPAR (Ours) | - | 88.50 | 83.27 | 89.74 | 89.56 | 89.43 | 87.95 | 84.01 | 89.55 | 91.65 | 90.28 |

### 4.4 ABLATION STUDY

• **Compoment Analysis.** In this experiment, we employ CLIP as the pre-trained model to extract textual and visual features. To evaluate the effectiveness of our approach, we compare it with the baseline VTB model, in which the textual module is replaced by local and global hypergraphs. All models are trained on the RAPv1 and PETA datasets to ensure fair and consistent evaluation. Table 4 summarizes the performance of each model in terms of accuracy (Acc), mean accuracy (mA), precision (Prec), recall (Rec), and F1-score.

• **Effect of Different HyperGraph Encoders.** Table 5 presents the results of using different hypergraph encoders for pedestrian attribute recognition on the RAPV1 dataset. We compared several hypergraph encoders based on the UniGNN (Huang & Yang, 2021) framework, including UniGIN, UniGCN, UniGAT, and the improved UniGCN2. Each encoder employs a different graph neural network approach—graph isomorphism network (UniGIN), graph convolutional network (UniGCN),

Table 2: Comparison with state-of-the-art methods on RAPv1 and RAPv2 datasets. The best and second best results are highlighted in **bold** and underlined, respectively. Missing values are marked with "-".

| Methods | Publish | RAPv2 | | | | | RAPv1 | | | | |
|---|---|---|---|---|---|---|---|---|---|---|---|
| | | mA | Acc | Prec | Rec | F1 | mA | Acc | Prec | Rec | F1 |
| SSCNet (Jia et al., 2021a) | ICCV21 | - | - | - | - | - | 82.77 | 68.37 | 75.05 | _87.49_ | 80.43 |
| CAS (Yang et al., 2021) | IJCV21 | - | - | - | - | - | 84.18 | 68.59 | 77.56 | 83.81 | 80.56 |
| IAA (Wu et al., 2022) | PR22 | 79.99 | 68.03 | 78.75 | 81.37 | 79.69 | 81.72 | 68.47 | 79.56 | 82.06 | 80.37 |
| DRFormer (Tang & Huang, 2022) | NC22 | - | - | - | - | - | 81.81 | 70.60 | 80.12 | 82.77 | 81.42 |
| VAC (Guo et al., 2022) | IJCV22 | 79.23 | 64.51 | 75.77 | 79.43 | 77.10 | 81.30 | 70.12 | _81.56_ | 81.51 | 81.54 |
| DAFL (Jia et al., 2022) | AAAI22 | 81.04 | 66.70 | 76.39 | 82.07 | 79.13 | 83.72 | 68.18 | 77.41 | 83.39 | 80.29 |
| VTB (Cheng et al., 2022) | TCSVT22 | 81.34 | 67.48 | 76.41 | 83.32 | 79.35 | 82.67 | 69.44 | 78.28 | 84.39 | 80.84 |
| PARformer (Fan et al., 2023) | TCSVT23 | - | - | - | - | - | 84.43 | 69.94 | 79.63 | **88.19** | 81.35 |
| OAGCN (Lu et al., 2023) | TMM23 | - | - | - | - | - | **87.83** | 69.32 | 78.32 | 87.29 | _82.56_ |
| SSPNet (Shen et al., 2024) | PR24 | - | - | - | - | - | 83.24 | 70.21 | 80.14 | 82.90 | 81.50 |
| SOFA (Wu et al., 2024) | AAAI24 | 81.9 | 68.6 | 78.0 | 83.1 | 80.2 | 83.40 | 70.00 | 80.00 | 83.00 | 81.20 |
| FRDL (Zhou et al., 2024) | ICML24 | - | - | - | - | - | _87.72_ | - | - | - | 79.16 |
| PromptPAR (Wang et al., 2024a) | TCSVT24 | _83.14_ | 69.62 | 77.42 | **85.73** | 81.00 | 85.45 | 71.61 | 79.64 | 86.05 | 82.38 |
| SequencePAR (Jin et al., 2025a) | PR25 | - | **70.14** | **81.37** | 81.22 | 81.10 | - | 71.47 | **82.40** | 82.09 | 82.05 |
| EVSITP (Wu et al., 2025a) | CVPR25 | **83.83** | 69.32 | 77.64 | 85.13 | 81.21 | 86.10 | _71.64_ | 79.24 | 86.65 | **82.78** |
| HDFL (Wu et al., 2025b) | Neural Networks25 | 81.81 | 68.22 | 78.30 | 82.18 | 79.85 | 83.68 | 70.49 | 80.25 | 83.55 | 81.51 |
| FOCUS (An et al., 2025) | ICME25 | - | - | - | - | - | 83.45 | 70.14 | 80.10 | 85.18 | 80.91 |
| KGPAR (Ours) | - | 83.02 | _70.03_ | _78.33_ | _85.17_ | **81.27** | 85.05 | **71.75** | 79.95 | 85.98 | 82.50 |

Table 3: Comparison with state-of-the-art methods on MSP60K datasets. The best and second best results are highlighted in **bold** and underlined, respectively. Missing values are marked with "-".

| Methods | Publish | Random Split | | | | | Cross-domain Split | | | | |
|---|---|---|---|---|---|---|---|---|---|---|---|
| | | mA | Acc | Prec | Recall | F1 | mA | Acc | Prec | Recall | F1 |
| DeepMAR (Li et al., 2015b) | ACPR15 | 70.46 | 72.83 | 84.71 | 81.46 | 83.06 | 54.84 | 44.97 | 63.38 | 58.81 | 61.01 |
| RethinkingPAR (Jia et al., 2021b) | arXiv20 | 74.01 | 74.20 | 84.17 | 83.94 | 84.06 | 55.98 | 46.52 | 62.85 | 62.09 | 62.47 |
| SSCNet (Jia et al., 2021a) | ICCV21 | 69.71 | 69.31 | 79.22 | 82.47 | 80.82 | 52.84 | 40.88 | 56.26 | 58.64 | 57.43 |
| VTB (Cheng et al., 2022) | TCSVT22 | 76.09 | 75.36 | 83.56 | 86.46 | 84.56 | 58.59 | 49.81 | 65.11 | 66.11 | 65.00 |
| Label2Label (Li et al., 2022) | ECCV22 | 73.61 | 72.66 | 81.79 | 84.32 | 82.56 | 56.38 | 45.81 | 59.67 | 64.20 | 61.19 |
| DFDT (Zheng et al., 2023) | EAAI22 | 74.19 | 76.35 | **85.03** | 86.35 | 85.69 | 57.85 | 49.97 | 65.34 | 66.18 | 65.76 |
| Zhou et al. (Zhou et al., 2023) | IJCAI23 | 73.07 | 68.76 | 78.38 | 82.10 | 80.20 | 54.26 | 41.91 | 56.23 | 60.11 | 58.11 |
| PARformer (Fan et al., 2023) | TCSVT23 | 76.14 | 76.67 | 84.77 | 86.93 | 85.44 | 57.96 | 50.63 | 62.28 | 71.04 | 65.82 |
| VTB-PLIP (Zuo et al., 2024) | arXiv23 | 73.90 | 73.16 | 82.01 | 84.82 | 82.93 | 56.30 | 46.77 | 61.20 | 64.47 | 62.18 |
| Rethink-PLIP (Zuo et al., 2024) | arXiv23 | 69.44 | 68.90 | 79.82 | 81.15 | 80.48 | 57.18 | 46.98 | 63.57 | 62.16 | 62.86 |
| PromptPAR (Wang et al., 2024a) | TCSVT24 | 78.81 | 76.53 | _84.40_ | 87.15 | 85.35 | 63.24 | 53.62 | **66.15** | 71.84 | 68.32 |
| SSPNet (Shen et al., 2024) | PR24 | 74.03 | 74.10 | 84.01 | 84.02 | 84.02 | 56.15 | 46.75 | 65.11 | 63.07 | 62.75 |
| HAP (Yuan et al., 2023) | NIPS24 | 76.92 | 76.12 | 84.78 | 86.14 | 85.45 | 58.70 | 50.59 | 65.60 | 66.91 | 66.25 |
| MambaPAR (Wang et al., 2024c) | arXiv24 | 73.85 | 73.64 | 83.19 | 84.29 | 83.28 | 56.75 | 47.34 | 61.92 | 64.98 | 62.80 |
| MaHDFT (Wang et al., 2024b) | arXiv24 | 74.08 | 74.40 | 82.82 | 86.41 | 83.93 | 58.07 | 50.65 | 62.39 | 71.13 | 65.85 |
| SequencePAR (Jin et al., 2025a) | PR25 | 71.88 | 71.99 | 83.24 | 82.29 | 82.29 | 57.88 | 50.27 | 65.81 | 65.79 | 65.37 |
| LLM-PAR (Jin et al., 2025b) | AAAI25 | **80.13** | **78.71** | 84.39 | **90.52** | **86.94** | **66.29** | **58.11** | 65.68 | **81.21** | **72.05** |
| KGPAR (Ours) | - | _79.03_ | _76.94_ | 84.14 | _88.14_ | 85.69 | _63.82_ | _53.67_ | 65.86 | _72.35_ | 68.41 |

graph attention mechanism (UniGAT), and the enhanced UniGCN2—demonstrating their effectiveness in pedestrian attribute recognition tasks.

## 4.5 VISUALIZATION

• **Heatmap Visualization.** To provide a more comprehensive illustration of the model's visual attention mechanism when predicting pedestrian attributes on the RAPV1 dataset, we adopt a heatmap-based visualization to highlight the critical response regions. As illustrated in Fig. 3, the model effectively attends to semantically relevant regions that are highly correlated with different attributes during inference, thereby demonstrating strong interpretability and a robust discriminative capability in attribute-specific feature localization.

Table 4: Ablation Study on RAPv1, and PETA datasets

| # | Baseline | Local-HG | Global-HG | RAPv1 | | | | | PETA | | | | |
|---|---|---|---|---|---|---|---|---|---|---|---|---|---|
| | | | | mA | Acc | Prec | Recall | F1 | mA | Acc | Prec | Recall | F1 |
| 1 | ✓ | | | 83.58 | 70.02 | 78.63 | 84.87 | 81.24 | 87.05 | 81.18 | 88.29 | 88.05 | 87.93 |
| 2 | ✓ | ✓ | | 85.28 | 71.19 | 79.51 | 85.66 | 82.11 | 88.54 | 82.02 | 88.65 | 89.04 | 88.60 |
| 3 | ✓ | | ✓ | 84.78 | 71.36 | 79.31 | 86.16 | 82.24 | 87.98 | 82.13 | 88.58 | 89.43 | 88.75 |
| 4 | ✓ | ✓ | ✓ | 85.05 | 71.75 | 79.95 | 85.98 | 82.50 | 88.50 | 83.27 | 89.74 | 89.56 | 89.43 |

Table 5: Comparison of different HyperGraph Encoders architectures on the RAPV1 dataset for pedestrian attribute recognition.

| KG Encoders | mA | Acc | Prec | Recall | F1 |
|---|---|---|---|---|---|
| UniGIN | 85.30 | 71.64 | 79.53 | 86.29 | 82.41 |
| UniGCN | 85.36 | 71.46 | 79.43 | 86.14 | 82.30 |
| UniGAT | 85.48 | 71.58 | 79.72 | 85.91 | 82.36 |
| UniGCN2 | 85.05 | 71.75 | 79.95 | 85.98 | 82.50 |

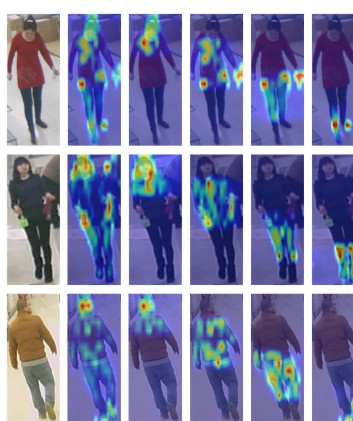
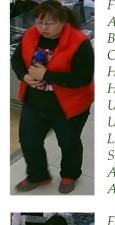
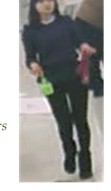
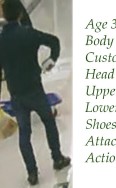
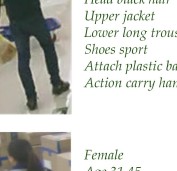
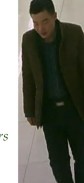
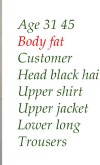
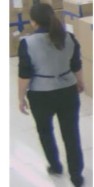

Figure 3: Visualization of heat maps given the corresponding pedestrian attribute and predicted attribute results.

• **Attributes Predicted.** As illustrated in Fig. 3, this work provides multiple examples of attribute recognition results on the RAPV1 dataset. Both quantitative and qualitative analyses indicate that the model achieves high stability and accuracy in identifying several key attribute categories, including gender, age, and clothing style. Even in complex scenarios, the model demonstrates consistent robustness in recognizing common attributes.

### 4.6 LIMITATION ANALYSIS

The model achieves competitive results but has some limitations. It relies on predefined body region partitions, which limits its ability to capture fine-grained attributes, particularly for accessories or complex textures. The fixed thresholds for constructing hypergraphs may not fully capture rare attributes, leading to suboptimal performance in such cases. Additionally, datasets like PETA and PA100K suffer from annotation inconsistencies and imbalances, which affect generalization. Lastly, the computational overhead from cross-modal hypergraphs may hinder large-scale real-time deployment. Future work will focus on adaptive partitioning, improved hypergraph construction, and lightweight variants for better efficiency.

## 5 CONCLUSION

We propose KGPAR, a hierarchical cross-modal hypergraph framework for pedestrian attribute recognition. By combining local and global hypergraph learning, it effectively captures intra-image feature interactions and inter-sample semantic relationships between visual and textual modalities. Extensive experiments on PETA, PA100K, RAPv1, RAPv2, and MSP60K demonstrate competitive performance across all metrics, showcasing improved robustness and balanced prediction. Future work will explore dynamic hypergraph construction and more efficient learning to enhance cross-dataset generalization and real-time applicability.

## ETHICS STATEMENT

This research does not involve human subjects, personally identifiable information, or sensitive data. All datasets used in this work are publicly available and widely used in the community. We have followed responsible research practices and ensured that no private or proprietary data was included. The proposed methodology is intended for academic research purposes and does not pose foreseeable risks of misuse. We believe our study complies with the ICLR Code of Ethics.

## REPRODUCIBILITY STATEMENT

To ensure the reproducibility of our work, we have included the complete implementation of our proposed method in the supplementary materials. The released code is accompanied by a detailed `README.md` file, which provides step-by-step instructions and execution commands for reproducing our experiments. This includes dataset preprocessing, model training, and evaluation procedures. With these resources, other researchers can readily verify and build upon our results.

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

## A  APPENDIX

• **Effect of Different Backbone Architectures.** Table 6 presents a comparative analysis of different backbone networks on the PA100k dataset. Replacing the ViT-B/16 backbone with the larger ViT-L/14 consistently improves all evaluation metrics, including mean accuracy (mA), overall accuracy (Acc), precision (Prec), recall, and F1 score. Specifically, ViT-L/14 achieves 87.95% mA and 90.28% F1, outperforming ViT-B/16 by 3.6% and 1.67%, respectively. This performance gain can be attributed to the greater representational capacity of ViT-L/14, which allows the model to better capture fine-grained pedestrian attributes. In contrast, the smaller ViT-B/16 backbone may underrepresent complex attribute interactions, leading to slightly lower precision and recall. Overall, these results highlight the importance of backbone selection in pedestrian attribute recognition, as larger models can more effectively encode discriminative features and enhance recognition robustness across diverse scenarios.

Table 6: Comparison of different backbone architectures on the PA100K dataset for pedestrian attribute recognition.

| Backbone | mA | Acc | Prec | Recall | F1 |
|---|---|---|---|---|---|
| ViT-B/16 | 84.35 | 81.45 | 88.16 | 89.79 | 88.61 |
| ViT-L/14 | 87.95 | 84.01 | 89.55 | 91.65 | 90.28 |

• **Effect of Different Number of Vision Context Samples.** Table 7 presents the effect of varying the number of images applied to each node's visual attributes on pedestrian attribute recognition performance using the RAPv1 dataset. During knowledge graph construction, a fixed number of images are assigned to each node's visual attributes. This experiment compares the impact of different image quantities on the performance of the method.

Table 7: Comparison of the effect of the number of images applied to each node in the knowledge graph on pedestrian attribute recognition performance on the RAPv1 dataset.

| number | mA | Acc | Prec | Recall | F1 |
|---|---|---|---|---|---|
| 2 | 85.22 | 71.60 | 79.86 | 85.79 | 82.39 |
| 4 | 85.09 | 71.48 | 79.28 | 86.31 | 82.30 |
| 6 | 84.93 | 71.26 | 79.47 | 85.76 | 82.14 |
| 8 | 84.81 | 71.09 | 79.01 | 86.08 | 82.04 |
| 10 | 85.05 | 71.75 | 79.95 | 85.98 | 82.50 |

• **Effect of Different Regional Partitioning Methods.** Table 8 presents a comparative analysis of regional versus non-regional partitioning strategies on the RAPv2 dataset. The regional division method achieves slightly higher mean accuracy (mA) and F1 score compared to the non-regional approach, with 83.02% mA and 81.27% F1. This indicates that explicitly dividing the image into semantic regions can help the model capture localized attribute features more effectively. However, the non-regional method exhibits marginally higher recall (85.37% vs. 85.17%), suggesting that global features may sometimes provide complementary information for certain attributes. Overall, incorporating regional partitioning improves the model's ability to focus on fine-grained attribute details while maintaining competitive overall performance, demonstrating its usefulness in pedestrian attribute recognition tasks.

Table 8: Comparison of regional and non-regional partitioning methods on the RAPv2 dataset for pedestrian attribute recognition.

| Regional Method | mA | Acc | Prec | Recall | F1 |
|---|---|---|---|---|---|
| Regional division | 83.02 | 70.03 | 78.33 | 85.17 | 81.27 |
| non Regional division | 83.37 | 69.50 | 77.53 | 85.37 | 80.91 |

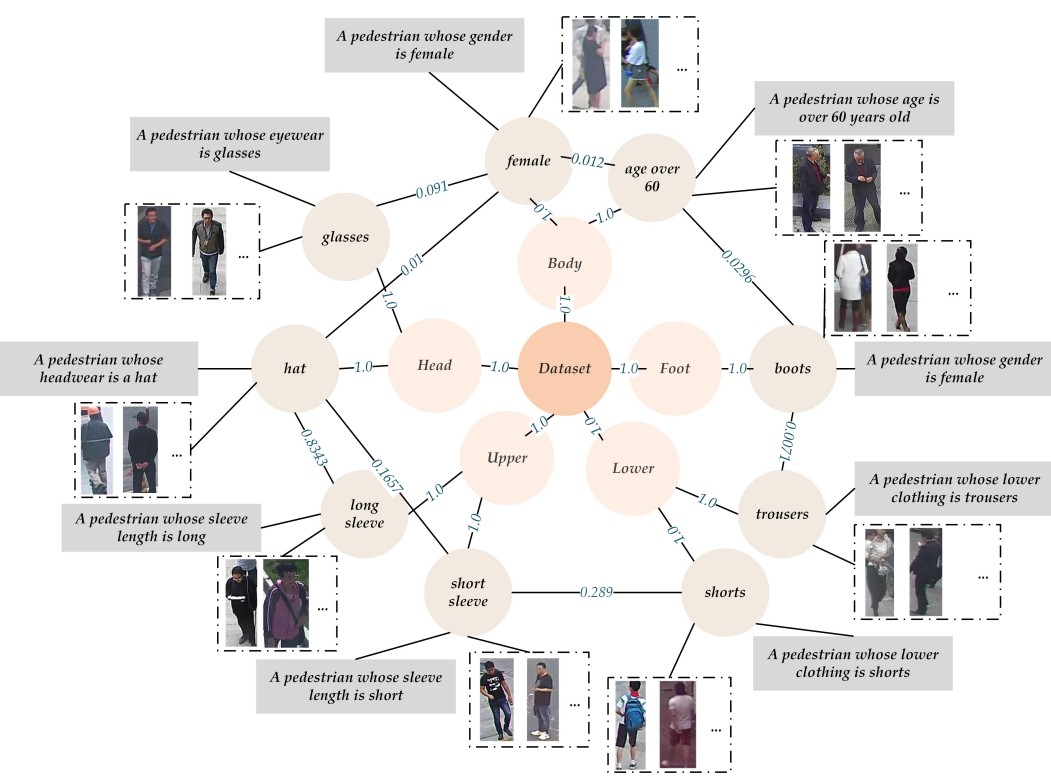

Figure 4: An overview of the built knowledge graph for pedestrian attribute recognition.

