# OpenReview forum: "Pedestrian Attribute Recognition via Hierarchical Cross-Modality HyperGraph Learning"
_ICLR.cc/2026/Conference — ICLR 2026 Conference Withdrawn Submission_

### Official Review · Reviewer_M6P4 · 2025-10-25

**Soundness:** 2
**Presentation:** 1
**Contribution:** 3
**Rating:** 4
**Confidence:** 4

**Summary:**

This paper presents a knowledge graph–guided hierarchical hypergraph framework (KGPAR) that effectively bridges visual and semantic modalities for pedestrian attribute recognition. The method is technically novel and empirically strong, achieving balanced gains across multiple benchmarks (Tables 1–3). However, it lacks computational analysis, relies on fixed structural choices, and provides limited ablation diversity, which leaves scalability and generalization insufficiently demonstrated.

**Strengths:**

The method combines local and global hypergraphs to capture higher-order vision-text relations, effectively bridging the semantic gap in prior PAR approaches. It constructs interpretable attribute nodes with textual and visual embeddings, enhancing semantic richness and supporting knowledge-guided learning. Experiments are conducted across multiple benchmarks, and heatmap visualizations provide insightful interpretations of the model’s reasoning.

**Weaknesses:**

See questions.

**Questions:**

- The performance comparisons across multiple benchmarks do not consistently achieve the best results. Additionally, Section 4.3 is poorly written, containing largely redundant content that occupies significant space; it should instead focus on analyzing the underlying reasons for the observed performance.

- The heatmap visualizations in Figure 3 are insightful for understanding attribute-level results, but they should be accompanied by qualitative comparisons with state-of-the-art methods.

- It is recommended to report training and inference time, GPU memory usage, and FLOPs for both the baseline and the full model in Section 4.2.

- Several important ablation studies are missing. Table 4 should be extended to include experiments on the loss coefficient α, different fusion strategies (concatenation vs. attention), and variations in graph depth.

---

### Official Review · Reviewer_cjTk · 2025-10-27

**Soundness:** 3
**Presentation:** 3
**Contribution:** 2
**Rating:** 6
**Confidence:** 3

**Summary:**

This paper proposes a novel multi-modal knowledge graph to model the connections between visual features, attribute texts, and broader visual contexts. It introduces a method for constructing this graph and a knowledge graph-guided cross-modal hypergraph learning framework to enhance standard PAR.

**Strengths:**

1. The paper introduces a hierarchical cross-modal hypergraph learning framework that effectively integrates visual and textual modalities through a multi-modal knowledge graph, offering a fresh perspective on semantic reasoning for PAR.
2. Extensive experiments on five benchmark datasets show competitive and balanced results across metrics, demonstrating the robustness and general applicability of the proposed model.
3. The model provides clear visualization and detailed ablation studies, supporting its design choices and offering interpretable insights into attribute–region relationships.

**Weaknesses:**

1. The proposed method relies on predefined body regions (e.g., head, upper, lower, foot), which cannot adapt to variations in pose, occlusion, or body proportions. This rigid division may lead to semantic misalignment between attributes and regions, such as misclassifying “hand accessories” as belonging to the “head” region when a pedestrian raises their hand.
2. Both the local and global hypergraphs are constructed using co-occurrence statistics and fixed similarity thresholds. Such a static design limits the model’s ability to capture rare or long-tail attributes; the model may underperform when handling uncommon attribute combinations or subtle semantic relations.
3. The M2PA-KG is built mainly from dataset co-occurrence and textual descriptions without incorporating external semantic priors or commonsense knowledge. So the “knowledge guidance” is largely statistical rather than semantic.
4. The paper does not provide a systematic study on key hyperparameters, such as the loss weight α (Eq. 13), similarity threshold τ (Eq. 15), or the number of visual samples per node. It is unclear how stable or robust the model’s performance is under different parameter settings.

**Questions:**

Please refer to ‘Weaknesses’.

---

### Official Review · Reviewer_Enh1 · 2025-11-03

**Soundness:** 3
**Presentation:** 2
**Contribution:** 2
**Rating:** 4
**Confidence:** 3

**Summary:**

This paper proposes a knowledge graph-guided hierarchical cross-modal hypergraph learning framework (KGPAR), which constructs a multi-modal pedestrian attribute knowledge graph (M2PA-KG) integrating visual and textual information to systematically model high-order relationships among pedestrian attributes and between attributes and visual features. Furthermore, it designs both local and global hypergraph modules to achieve joint modeling of fine-grained and global semantics, thereby significantly improving the accuracy of pedestrian attribute recognition.

**Strengths:**

1. This method introduces textual information and establishes local–global mapping relationships between text and visual features, enhancing the fusion and utilization of multimodal information.

2. Experiments conducted on five datasets validate the effectiveness and generalization capability of the proposed approach.

3. The visualization heatmaps enable readers to intuitively perceive the correspondence between pedestrian attributes and image regions.

**Weaknesses:**

1. The performance on the PETA and PA100K datasets is worrying, and on the MSP60K dataset, it is significantly worse than the LLM-PAR method. Furthermore, the comparison method on the MSP60K dataset is inconsistent with other datasets, which is not explained.

2. Whether fine-tuning is needed when using CLIP for local alignment is not mentioned. Whether CLIP can be directly applied to the alignment of fine-grained text descriptions and local images is also not explained.

3. The threshold in Local HyperGraph has not undergone ablation experiments, and no explanation or interpretability is provided for the threshold selection. This method relies on predefined body regions for local mapping, and alignment errors are inevitable due to different threshold choices. However, the paper does not explain whether such errors lead to a performance degradation, or by how much. Therefore, ablation experiments on the threshold are crucial.

4. The $\alpha$ parameter in Equation 13 has not undergone ablation experiments, and even its specific value is not explained.

**Questions:**

Please refer to "Weaknesses".

---

### Note · Authors · 2025-11-12

I have read and agree with the venue's withdrawal policy on behalf of myself and my co-authors.